# Regulation of Molecular Targets in Osteosarcoma Treatment

**DOI:** 10.3390/ijms232012583

**Published:** 2022-10-20

**Authors:** Betul Celik, Kader Cicek, Andrés Felipe Leal, Shunji Tomatsu

**Affiliations:** 1Department of Biological Science, University of Delaware, Newark, DE 19716, USA; 2Nemours/Alfred I. DuPont Hospital for Children, Wilmington, DE 19803, USA

**Keywords:** gene silencing and knockdown, lentiviral vectors, osteosarcoma, RNAi, shRNA, siRNA, miRNA

## Abstract

The most prevalent malignant bone tumor, osteosarcoma, affects the growth plates of long bones in adolescents and young adults. Standard chemotherapeutic methods showed poor response rates in patients with recurrent and metastatic phases. Therefore, it is critical to develop novel and efficient targeted therapies to address relapse cases. In this regard, RNA interference technologies are encouraging options in cancer treatment, in which small interfering RNAs regulate the gene expression following RNA interference pathways. The determination of target tissue is as important as the selection of tissue-specific promoters. Moreover, small interfering RNAs should be delivered effectively into the cytoplasm. Lentiviral vectors could encapsulate and deliver the desired gene into the cell and integrate it into the genome, providing long-term regulation of targeted genes. Silencing overexpressed genes promote the tumor cells to lose invasiveness, prevents their proliferation, and triggers their apoptosis. The uniqueness of cancer cells among patients requires novel therapeutic methods that treat patients based on their unique mutations. Several studies showed the effectiveness of different approaches such as microRNA, drug- or chemotherapy-related methods in treating the disease; however, identifying various targets was challenging to understanding disease progression. In this regard, the patient-specific abnormal gene might be targeted using genomics and molecular advancements such as RNA interference approaches. Here, we review potential therapeutic targets for the RNA interference approach, which is applicable as a therapeutic option for osteosarcoma patients, and we point out how the small interfering RNA method becomes a promising approach for the unmet challenge.

## 1. Introduction

Bone cancer irregulates cell growth in the bone. It can come in many forms depending on the type of bone cell transformed. Osteosarcoma is the eighth most common childhood malignancy, comprising 2.4% of pediatric cancers, including leukemia (30%), brain and nervous system cancers (22.3%), neuroblastoma (7.3%), Wilms tumor (5.6%), non-Hodgkin lymphoma (4.5%), rhabdomyosarcoma (3.1%), retinoblastoma (2.8%), and Ewing sarcoma (1.4%) [1,2]. Osteosarcoma, which has a bimodal age distribution, results from the cells mutated in osteoblastic lineage depending on their susceptibility during osteoblastic differentiation [1]. Osteosarcoma cells from osteoblastic lineage induce mesenchymal bone marrow cells into cancer-associated cells [3]. Osteosarcoma affects the pediatric group (mostly 10–14 years old age) accounting for the first osteosarcoma peak and the older adulthood group (older than 65 years of age) for the second osteosarcoma peak [2]. The incidence of osteosarcoma is higher in males (5.4 per million persons per year) than in females (4.0 per million persons per year). Among populations, black people have the highest incidence of osteosarcoma (6.8 per million persons per year), followed by Hispanics (6.5 per million) and Whites (4.6 per million). The most common site of osteosarcoma is commonly in the long bones near the metaphyseal growth plates; the femur (42%; 75% of tumors in the distal femur), the tibia (19%; 80% of tumors in the proximal tibia), and the humerus (10%; 90% of tumors in the proximal humerus). Additionally, osteosarcoma may occur in the skull or jaw (8%) and the pelvis (8%). Regarding mortality rates, bone and joint malignancies accounted for 8.9% of all childhood and adolescent cancer deaths [2].

Studies have shown that somatic copy number changes, along with recurrent point mutations, often lead to the development of osteosarcoma [4,5]. In recent years, the prevalence of germline mutations among pediatric cancer patients has reached 7.9%, which is associated with several cancer predisposition disorders such as autosomal dominant Li-Fraumeni and hereditary retinoblastoma, and autosomal recessive Werner, Bloom, Rothmud–Thompson, and Rapadilino syndromes [6]. Taking all into consideration, it is important to improve an effective treatment for osteosarcoma. Chemotherapy, radiotherapy, and surgical resections have been widely used to treat bone tumors; however, these techniques have some disadvantages. For instance, they cannot quickly determine the unknown vulnerabilities, such as chaotic chromosomal rearrangements causing the amplification of oncogenes or loss of tumor suppressors [7], in osteosarcoma tumors [8]. On the other hand, gene-silencing approaches appear to be more promising therapies because of the potential to target a specific gene to up or downregulate permanently. Gene regulation approaches under gene therapy are considered an effective, safe, fast, and one-time treatment method to prevent the poor prognosis of cancer and other diseases compared to chemotherapeutic methods. In vitro experiments have demonstrated that silencing the protein-coding genes decreases messenger RNA (mRNA) production and protein expression while inducing apoptosis in osteosarcoma cells [9]. Furthermore, in vivo experiment results have revealed that gene-silencing reduced cell proliferation and tumor growth in a murine osteosarcoma model [10]. The genome-wide sequence studies show that the genome of osteosarcoma is complicated since the genome profiles differ significantly among osteosarcoma patients [5]. This finding causes a significant obstacle to identifying the etiology of osteosarcoma and interferes with the development of effective treatment. Unlike synovial and Ewing sarcomas, characterized by chromosomal translocation, osteosarcoma accounts for various molecular alterations and cytogenetic instability. Thus, identifying prognostic and genetic markers and drivers is challenging to develop targeted therapies [11]. Therefore, genome-informed targeted therapies require more experiments to improve their therapeutic efficacy in osteosarcoma and to establish more personalized therapies.

To date, ribonucleic acid interference (RNAi) is a promising approach to silence overexpressing genes, which directly affects mRNA degradation pathways (Figure 1) [9]. In the RNAi approach, long mRNA transcripts are diced by the Dicer enzyme to create short double-stranded RNAs (dsRNAs), and these dsRNAs are loaded onto RNA-induced silencing complex (RISC) comprising Argonaute and transactivation response RNA-binding protein. After cleaving one strand, the guide strand is paired with its complementary mRNA target via RISC. Following binding, the mRNA is silenced via two pathways: RNase-mediated degradation or translational repression [12,13,14,15]. RNAi approach has four types: small interfering RNA (siRNA) and microRNA (miRNA) mimics, short hairpin RNAs (shRNAs), and Dicer substrate RNAs (dsiRNAs), which are non-coding RNAs (ncRNAs) and post-transcriptionally regulate protein synthesis. siRNA is a short double-stranded ncRNA, having 20–25 nucleotides loaded onto RISC, and they degrade and cleave mRNAs containing specific nucleotide sequences before being translated. miRNA having 19–25 nucleotides is another double-stranded ncRNA loaded onto RISC to regulate gene at the posttranscriptional level by targeting mRNA sequence [16,17]. shRNA requires nuclear processing and is most upstream, while dsiRNA requires Dicer processing. siRNA and miRNA pathways are the most direct; however, their silencing outcome is different because siRNAs are 100% complementary of mRNA target sequences, but miRNAs are not. miRNAs induce translational repression, while siRNAs induce Argonaute2-mediated degradation [12]. All four types of RNAi approaches are delivered by inserting into either nanoparticles or viral vectors or alone currently in clinical trials [12]. As a part of the natural cellular process, siRNAs are administered directly into cells, tissue, or organisms; however, shRNAs that provide siRNA are transported into cells through a vector, and siRNA is expressed when shRNA is processed [18]. Murine experiments demonstrated that viral vectors could be potential therapeutic agents for treating genetically acquired diseases [19] or diseases transmitted by genetic predisposition [10,20,21].

Here, we mainly focus on the lentiviral gene delivery RNAi approach and its effectiveness in potential molecular targets of osteosarcoma. This review aims to compare molecular targets and to show the effectiveness of gene-silencing therapy in osteosarcoma treatment.

## 2. Molecular Targets in Osteosarcoma Treatment

### 2.1. Ubiquitin-Specific Protease 1

Dysfunction and dysregulation of the ubiquitin–proteasome system affect nearly all vital cellular processes, from gene transcription and DNA repair to cell cycle regulation and apoptosis [22], and profoundly induce tumor progression [23]. Ubiquitination and deubiquitination, the most important posttranslational modifications, regulate the metabolic reprogramming in cancer cells [24]. Ubiquitin-specific protease 1 (USP1) is a subtype of deubiquitinating enzymes involved in DNA damage and repair the response by deubiquitinating the proliferating cell nuclear antigen, Fanconi anemia group D2 and group 1 [23]. Studies have shown that USP1 is highly expressed in malignant tumors including osteosarcoma, colorectal cancer, and non-small cell lung cancer (NSCLC) [9,25,26]. Liu et al. (2016) described 30 osteosarcoma patient samples and the commercial human osteosarcoma cell line (U2OS) stained for USP1 expression. The results confirmed that USP1 was overexpressed in 26 samples, and the highest USP1 expression was demonstrated in cartilage tumor tissues and osteosarcoma tissues compared to normal bone tissues. To validate the role of USP1 overexpression in osteosarcoma, U2OS cells were transduced with a lentiviral vector carrying a USP1-silencing siRNA sequence. Real-time PCR and Western blot analysis showed that USP1-siRNA transduction effectively reduced both USP1 mRNA and protein expression levels, and the knockdown of USP1 downregulated the expression of SIK2, MMP-2, GSK-3β, Bcl-2, Stat3, cyclin E1, Notch1, Wnt-1 and cyclin A1, most of which contribute to the tumor growth and development. Furthermore, the silencing of USP1 inhibited tumor growth and colony-forming while reducing the invasiveness of U2OS cells. USP1-siRNA-transduced U2OS cells exhibited decreased cell viability as gene silencing, resulting in reduced colony formation and increased apoptosis. USP1 gene silencing also prevented U2OS invasion [9].

Salt-inducible kinase 2 (SIK2) played critical roles in bipolar spindle formation and cAMP response element-binding protein (CREB) mediated gene transcription. Overexpression of SIK2 on several tumors, including osteosarcoma, suggested a potential role in cancer development [27]. SIK2 also regulates mitotic progression and transcription in prostate cancer [28]. Consequently, the relationship between USP1 and SIK2 was investigated. The silencing of USP1 also inhibited SIK2 expression, while using MG132 (proteasome inhibitor) led to its increase. Silencing SIK2 by the lentiviral vector-delivered USP1-shRNA resulted in increased apoptosis and poor invasiveness of U20S cells, supporting their use as a potential therapeutic target [9]. Furthermore, Williams et al. showed that USP1 regulates mesenchymal stem cells (MSCs) and osteosarcoma cells by deubiquitinating the inhibitors of DNA binding (IDs), which maintain and regulate stem cell differentiation [29]. Transcriptionally induced IDs have four types; ID1, ID2, ID3, and ID4, which antagonize the basic helix-loop-helix proteins. Bone morphogenic proteins, platelet-derived growth factor, epidermal growth factor, and T cell receptors ligation are inducers of IDs [30]. Following MG132 treatment, ID2 accumulation was shown in 293T cells. To identify the effect of USP1 on ID2, 293T cells were transfected with USP1-carrying plasmid and analyzed by ID2 abundance. Results showed that while the half-life of ID2 in the cells transfected with the control vector was about 2 min, overexpressed USP1 increased the half-life of ID2 to approximately 80 min, which validates that USP1 stabilizes ID2. Then, they validated osteoblastic differentiation in the USP1 knockdown osteosarcoma xenograft model. USP1 knockdown reduced ID1 and ID2, whereas it promoted the expression of differentiation markers, including osteonectin, RUNX2, SPP1/osteopontin, osterix, and BGLAP/osteocalcin [29].

Moreover, a recent study has revealed another role of USP1 in OS: that USP1 mediates the stabilization of transcriptional co-activator with PDZ-binding motif (TAZ) via K11 and K29 ubiquitylation, which affects the downstream Hippo signaling pathway. In this study, the researcher identified that the depletion of USP1 resulted in diminished TAZ translocation into the nucleus. This was validated by co-immunoprecipitation, showing that USP1 depletion inhibits the interaction between TEAD transcription factors and TAZ in OS cells. In addition, USP1 depletion also reduced the downstream components of the Hippo signaling pathway level, such as CYR61, c-Myc, and RUNX2 [23].

There are some other deubiquitinating genes targeted as those in USP1. A study showed that ubiquitin carboxyl-terminal hydrolase L1 (UCHL1), a subtype of deubiquitinases, could act as an oncogene, which was elevated in osteosarcoma cells compared to healthy bone tissues. The expression level of UCHL1 influenced tumor size, high lung metastasis rate, and short survival time. Silencing of UCHL1 by lentiviral vector led to the inhibition of cell proliferation and increased cell population in the G1 phase. Additionally, the knockdown of UCHL1 reduced cyclin D1, cyclin E1, and CDK6 promoting G1/S phase transition, inhibiting cell invasion, and inducing cell apoptosis. The correlation found between UCHL1 and Akt/ERK signaling pathway pointed out that UCHL1 accelerated the osteosarcoma progression mediating the Akt signaling pathway [31].

BRCA1-associated protein 1 (BAP1) is another nuclear-localized deubiquitinating enzyme, the dysregulation of which was unknown in osteosarcoma. A study reported a reduced amount of BAP1 in 30 osteosarcoma patients compared to control subjects. To investigate the biological function and molecular mechanisms of BAP1, MG-63 and SJSA-1 osteosarcoma cell lines were transiently transduced with a lentiviral vector to overexpress BAP1 and then stably transfected with the plasmid carrying sgRNA by CRISPR-Cas9 to knockdown BAP1. The overexpression of BAP1 reduced the proliferation of MG63 and SJSA-1 cells, whereas the silencing of BAP1 promoted the proliferation of MG63 and SJSA-1 cells. As a result, it was illustrated that BAP1, by inhibiting phosphoinositide 3-kinase/Akt signaling, suppressed cell proliferation, apoptosis, migration, and invasion of osteosarcoma cells [32].

Ubiquitin-specific peptidase 7 (USP7) is another critical member of the deubiquitinating enzyme family. In a study, a high level of USP7 was reported in 45 osteosarcoma samples, and it induced epithelial-mesenchymal transition (EMT) in osteosarcoma cells by activating the Wnt/β-catenin signaling pathway [33].

Taken all together, deubiquitinases might be potential diagnostic and therapeutic targets in osteosarcoma treatment.

### 2.2. ErbB Receptor Family

Erythroblastic leukemia oncogene homolog (ErbB) receptors are a family of receptor tyrosine kinases responsible for regulating critical cell signaling pathways and gene expression levels [34]. They also play an essential role in developing and progressing many cancer types [10,35]. Some of these receptors are in the nucleus, where many oncogenes can be upregulated, including cyclinD1, B-myb, cyclooxygenase-2, and iNOS/NO [36]. There are four members of the ErbB family: ErbB1 [epidermal growth factor receptor (EGFR) HER1], ErbB2 (HER2), ErbB3 (HER3), and ErbB4 (HER4) [36].

ErbB1 and ErbB2 promote tumor development, while ErbB2/ErbB3-signaling promotes cell growth and tumor cell invasion via the PI3K/Akt pathway. However, the role of these receptors in soft tissue tumors and sarcomas remains unclear [21,37,38]. N. Jullien et al. reported that ErbB3 expression is significantly higher in U2OS, MG63, and SaOS2 osteosarcoma cell lines than in normal primary osteoblast cell lines (N976 and N704). ErbB3 protein levels in human osteosarcoma cells were higher than those in normal human primary osteoblast cells [21]. Furthermore, higher expression of ErbB3 was associated with increased metastases and recurrent disease. To silence the ErbB3 expression, a pLKO.1-anti-ErbB3 shRNA (shErbB3) was designed and added into lentiviral vectors. K7M2 cells (aggressive murine osteosarcoma cells) were transduced to determine the role of ErbB3 in osteosarcoma cell growth [10]. Knockdown of ErbB3 resulted in a significant decrease in the proliferation ratio in K7M2 cells, although apoptosis did not increase. K7M2 are apoptotic-resistant cells with enhancer aldehyde dehydrogenase (ALDH) activity, a well-known cancer stem cell marker, which confer, among others, resistance to apoptosis-mediated cell death [39]. Despite the above, the silencing of ErbB3 expression inhibited the invasion and migration of both metastatic K7M2 cells and bone tumor cells treated in this study [10]. Detailed studies on the tumoral microenvironment should be addressed to know the therapeutic impact of these strategies, given the persistence of these tumoral cells.

Another study has revealed that Wnt3a downregulates ErbB3, and this causes Wnt-induced osteoblast differentiation in MSCs [21]. MSCs differentiate between bone-forming osteoblasts and cartilage-forming chondrocytes, promoting the healing process. BMP, Wnt, and Notch signaling pathways affect MSC differentiation and proliferation [40]. Targeting of Wnt signaling in MSCs induces MSC osteoblast differentiation for bone regeneration. The study indicated that neuregulin 1 (NRG1), a member of the epidermal growth factor family of receptor tyrosine kinase, has been upregulated by Wnt3a signaling when osteoblast differentiation is induced in primary human MSC and murine model (C3H10T1/2) MSCs. Furthermore, NRG1 did not affect alkaline phosphatase (ALP) activity, indicating osteosarcoma differentiation. To determine the gene expression profile in both murine C3H10T1/2 cells and human mesenchymal cells, lentiviral vectors carrying pLKO.1-anti-ErbB3 shRNA (shErbB3) with or without 15% Wnt3a conditioned medium were adopted. The results showed that Wnt3a signaling increased NRG 1 and ET 1 mRNA levels in both hMSCs and murine pluripotent mesenchymal C3H10T1/2 cells, suggesting that these two genes are similarly targeted by Wnt signaling. Moreover, shErbB3 and Wnt3a increased ß-catenin transcriptional and ALP activities in MSC. The study also investigated the interaction between Src and ErbB3 signaling, in which Src proteins controlled the differentiation of Wnt3a-induced osteoblasts; when the ErbB3 gene was silenced, Src levels increased [21]. Additionally, the neuroglin/ErbB3 signaling was constitutively activated in clear cell sarcoma of soft tissues, in which when ErbB3 lost the kinase activity, ErbB2 was induced with an autocrine stimulation [21].

Huang et al. (2019) investigated ErbB receptor family amplification in primary osteosarcoma and their correlation with clinicopathological and prognostic values [38]. Ninety samples were collected through surgical resections from primary osteosarcoma patients, thirty of which were non-neoplastic bone tissues. Western blot and reverse transcription–quantitative polymerase chain reaction (RT-qPCR) analyses verified EGFR protein and mRNA overexpression and higher amplification levels of EGFR, ErbB3, and ErbB4 in osteosarcoma tissues compared to non-neoplastic tissues. Another study result was that the elevated ErbB3 signaling and ErbB3-EGFR co-amplification levels in tumorigenesis, tumor progression, and drug resistance were associated with poor chemotherapy response, distant metastasis, and poor progression, which further verified that ErbB3 is a significant therapeutic target in osteosarcoma treatment [38].

Many in vivo studies have been performed to evaluate gene-silencing effects on osteosarcoma growth. In a murine allograft model, murine K7M2 cells were transduced with an ErB3 silencing shRNA and placed into BALB/C mice [10]. The analysis revealed that ErbB3 silencing dramatically decreased the number and size of tumors, but increased apoptosis was not found. This indicates that ErbB3 silencing affects cell number by inhibiting cell proliferation in mice [10].

### 2.3. Lysophosphatidic Acid Acyltransferase ß (LPAATβ)

LPAATβ, an enzyme converting intracellular lysophosphatidic acid (LPA) to phosphatidic acid (PA), regulates osteosarcoma cell proliferation. The studies revealed that LPA stimulates cell proliferation, migration, and survival. PA is another biologically active phospholipid that affects most signal transduction pathways, such as mTOR and Raf-1. LPAATβ is currently the most preferred target for osteosarcoma research [41,42]. In research, Song et al. (2017) examined 40 osteosarcoma patients aged 13 to 46 years to detect the expression level of LPAATβ and other related proteins and studied whether the silencing of LPAATβ expression has an impact on osteosarcoma with cisplatin resistance. Furthermore, for further experiments, cisplatin-resistant samples were chosen by real-time polymerase chain reaction (RT-PCR) and Western blotting. siRNA silencing LPAATβ is inserted into the lentiviral vector and then administered to the cisplatin-resistant osteosarcoma cells both in vitro to detect the effect of this specific enzyme on cell viability and in vivo to detect tumor growth with cisplatin treatment. The results indicated that cisplatin-resistant sensitivity decreased when the PI3K/Akt/mTOR signaling pathway was activated. Moreover, tumor growth was inhibited by silencing LPAATβ in the xenografts nude mouse model with cisplatin-resistant osteosarcoma cells [42]. Most studies showed that abnormal activation of this signaling pathway causes the cells to be transformed into the malignant type and be resistant to chemotherapy by regulating multidrug resistance gene 1/P-glycoprotein (MDR1/P-gp) [42]. For instance, Ma et al. indicated that rapamycin-mediated mTOR inhibition reversed MDR in the colorectal cancer cell. Therefore, autophagy and apoptosis increased, whereas the expression of MDR1 was reduced in rapamycin-resistant cells treated with adriamycin [43]. A study implied that cytotoxic drugs led to cytotoxic stress; thus, the Akt signaling pathway was downregulated. This led to the hypophosphorylation of translational repressor 4E-BP (eukaryotic initiation factor) and the decrease in eIF4E availability. The downregulation of eIF4E due to cytotoxic stress decreased the translation efficiency of the MDR1 mRNA -structure [44]. Another study mentioned that the inhibition of PI3K by PI103, PI3K/mTOR inhibitor, enhances doxorubicin (DOX)-induced apoptosis with the help of the effective interaction of PI103 with DOX in sarcoma cells. This interaction activated the proapoptotic protein Bax, a well-known mitochondrial-dependent apoptosis trigger [45]. Overall, the activation of the PI3K/Akt/mTOR pathway has inhibited cisplatin-induced apoptosis and improved cancer cells to become cisplatin-resistant [42,46].

The role of LPAATβ has been investigated in 158 human ovarian cancer cases, of which 68 cases were in an advanced stage, by immunohistochemistry (IHC) for the LPAATβ expression in the local tumor and distant metastases. Ninety out of all one hundred and fifty-eight ovarian tumors overexpressed LPAATβ. Out of 68 advanced-stage tumors, 49 overexpressed LPAATβ [47]. Thirty-three out of one hundred and fifty-eight samples showed highly aggressive histology depending upon LPAATβ expression, in which progression-free survival (PFS) and overall survival decreased [47]. Moreover, gynecologic research illustrated that LPAATβ was elevated in gynecologic malignancies, associated with reduced survival in ovarian cancer and earlier disease progression in ovarian and endometrial cancer. However, inhibiting LPAATβ by siRNA or selective inhibitors (CT32521 and CT32228) induces apoptosis in human ovarian and endometrial cancer cell lines. Apart from that, LPAATβ inhibition, by either siRNA or these inhibitors, increased the survival of mice bearing ovarian tumor xenografts [48].

### 2.4. Notch Signaling Pathway

The Notch signaling pathway regulates cell-to-cell signal transduction, in which the signal is produced when Notch signal-related ligands of adjacent cells bind with the receptors in the receiver cell. The ligand–receptor interaction makes conformational changes in the notch protein in the signal-receiving cell. Then, the Notch intracellular domain (NCID) is cleaved from the cellular membrane and released to the receiving cell nucleus. This translocation induces transcription of Notch target genes [49,50,51]. There are five Notch ligands (Jagged 1 and 2, Delta-like (DLL) 1, 3, and 4) and four Notch receptors (Notch 1-4) mainly composed of mammalian Notch signaling (Figure 2). Notch signaling has pivotal roles in cell fate, proliferation, apoptosis, and cell migration [51]. Notch signaling affects skeletal development and homeostasis. Mutations in Notch genes are associated with a selected group of genetic skeletal disorders, and dysregulated Notch signaling occurs in osteosarcoma and osteoarthritis. Modeling these diseases in mice has provided new insight into disease pathogenesis and established the importance of Notch signaling in skeletal development and function [52].

Cao et al. (2017) showed that the Notch signaling pathway promoted osteogenic differentiation and proliferation of mesenchymal stem cells by inducing bone morphogenic protein 9/Smad (BMP9/Smad) signaling and by upregulating ALK2 expression, respectively [51]. Furthermore, this study focused on the Notch signaling effects on BMP9-induced early and late osteogenic differentiation in MSCs. In early osteogenic differentiation of MSCs, the Notch activity of MSCs was inhibited with γ secretase inhibitor (DAPT) and adenoviral vectors carrying dnNotch1 (dominant-negative mutant of Notch 1) to downregulate and DLL1 (Delta-like receptor 1) to upregulate the Notch signaling were used. The results revealed that BMP9-induced alkaline phosphatase activity (ALP) was significantly inhibited by DAPT and Ad-dnNotch1, while Ad-DLL1 enhanced BMP9-induced ALP activity, which was evident that the Notch signaling enhanced the BMP9-induced early osteogenic differentiation of MSCs. ALP activity is a critical osteogenic marker at early osteogenic differentiation and bone formation; however, osteopontin (OPN) and osteocalcin (OCN) markers are critical for the late stage of osteogenic processes [51,55]. Therefore, the effects of Notch signaling on the expression of BMP9-induced late osteogenic markers were investigated. In this regard, the combination therapy with Ad-BMP9 and DAPT considerably decreased the OCN expression. In addition to that, Ad-dnNotch1 treatment decreased the expression of OCN and OPN in MSCs. On the contrary, treatment with Ad-DLL1 enhanced the matrix mineralization induced by BMP9, which proved that the Notch signaling promoted BMP9-induced late osteogenic differentiation [51].

Another study showed that Notch signaling increased BMP9-induced ectopic bone formation in 4-week-old athymic nude mouse models. The Notch signaling had a pivotal role in enhancing the BMP/Smad signal transduction induced by BMP9 in MSCs while regulating BMP9-induced osteogenic factors depending on different receptor/ligand responses. That is why the expression of activin receptor-like kinase 1 (ALK1) and activin receptor-like kinase 2 (ALK2) induced by BMP9 was experimented with MSCs to detect the Notch signaling influences. ALK2 was downregulated by Ad-notch1 combined with BMP9-conditioning media (BMP9-CM) in mouse embryonic fibroblasts. Ad-DLL1 combined with BMP9-CM also increased the percentage of mouse embryonic fibroblasts in the S phase, compared to Ad-RFP in combination with BMP9-CM. The percentage of cells in the G0/G1 phase decreased significantly, which indicated that the proliferation of MSCs may be facilitated in the presence of BMP9-CM [51]. BMP9 has critical roles in liver fibrosis [56], iron metabolism [57], cartilage formation [58] and angiopoiesis [59,60]. BMP9 is more potent in inducing osteogenesis and chondrogenesis of MSCs than other BMPs [58,61]. Moreover, BMP9 elevated the expression of type II collagen (COL2A1) mRNA and increased the expression of aggrecan and cartilage oligomeric matrix proteins [61].

The Notch signaling activation inhibited cell differentiation and resulted in bone osteopenia in the osteoblastic lineage. The studies showed that Notch1 inhibited osteoclastogenesis and bone resorption while Notch2 enhanced [52]. Increased notch expression (Notch1-2), higher levels of DLL1 mRNA and the Notch target gene HES1, and loss of p53 protein were associated with osteosarcoma, and the long-term activation of the Notch pathway in osteoblasts led to osteosarcoma in the mouse model [62,63]. The expression of Notch1 in newly formed osteoblast was sufficient to drive the formation of bone tumors, including osteosarcoma, which was accelerated in the mouse model when Notch activation was combined synergistically by the loss of p53. The experiments to show this synergy between Notch and p53 concluded that the Notch gain-of-function mutation promoted the progression of osteosarcoma induced by the loss of p53. This research also highlighted that Notch oncogene and p53 mutation had a dual dominance in osteosarcoma development. The activation of Notch signaling at any stage could induce the proliferation of immature osteoblasts while inhibiting their differentiation into mature osteoblasts [63].

### 2.5. Extracellular Matrix Molecules

The dysregulation and abnormal remodeling of the extracellular matrix (ECM) are notable for the disease progression and healing process, and it has gained prominence recently for osteosarcoma and other cancer types [64]. For most cancers, the tumor microenvironment (TME), which comprises blood vessels, fibroblasts, immune and endothelial cells, signaling molecules, extracellular vesicles, and mostly the ECM, has been found to affect the progression and metastasis [65]. The ECM can prevent cancer initiation at the early stages. In addition to that, it drives disease progression toward malignancy. The studies showed that the composition of the extracellular TME was growing evidence for the detection of clinical prognosis [66]. Bergamaschi et al. (2008) analyzed the matrix composition of 28 primary breast carcinomas regarding the morphology and differential expression of ECM-related gene profiles [67]. According to the results, 278 ECM-related genes and their expression profile were examined, in which ECM was classified into four main branches (ECM 1-4). Of twenty-eight samples, eight were grouped into ECM 1, which was associated with the PI3K pathway, lymphoid infiltration, and the upregulation of adhesion molecules and collagens; eight were involved in ECM 2, in which glucose metabolic pathways were overexpressed, and the hyaluronan was found to have a high level of expression among the extracellular protein-coding genes. Seven accounted for ECM 3 because of the enriched gene profile regulating Wnt-β-catenin pathways and membrane integrins, which had roles in connective tissue maintenance. Finally, five samples were included in ECM 4 tumors, where the genes involved in endoplasmic reticulum pathways and inflammation were upregulated. All ECM 1 and ECM 2 groups showed poor outcomes, while ECM 3 and 4 were less aggressive [67].

ECM comprises collagens, fibronectin, laminins, and proteoglycans, including biglycan, decorin, lumican, versican, and hyaluronan [68]. The studies indicated that collagen (type I, III, IV, V, and XVIII) affected invasion, metastasis, chemotherapy resistance, angiogenesis, adhesion, anti-angiogenesis, and cell growth in osteosarcoma. While the expression of collagen type XVIII diminished, the expression of other collagens elevated in osteosarcoma. Moreover, fibronectin and laminin expression in osteosarcoma increased. Laminins provided osteosarcoma cells to be adhesive and invasive, while fibronectins made osteosarcoma cells resistant to chemotherapy and rolled to metastasis, adhesion, and invasion [65]. Proteoglycans (PGs) are critical ECM components containing glycosaminoglycan (GAG) chains attached to the protein core. The GAGs play essential roles in cell signaling, modulating several biological processes such as cell growth and proliferation, adhesion, anticoagulation, and wound repair. Depending upon the core protein to which the GAG is bound, there are four types of GAGs: heparin/heparan sulfate (HS), chondroitin sulfate (CS), dermatan sulfate (DS), keratan sulfate (KS), and hyaluronic acid (HA) [69]. Sasisekharan et al. mentioned that HS has both tumorigenic in which it regulates autocrine signaling loops causing unregulated cell growth and anti-tumorigenic effects; HS facilitates the immune response against the growing tumor as a protective barrier and, in contrast, develops new blood vessels around the growing tumor depending on the location and sequence [70]. HSPGs have been categorized into two categories: cell-surface PGs, including syndecans, glypicans, and basement membrane PGs containing perlecan, agrin, and collagen type XVIII. The presence of HSPGs promotes cell adhesion, while their absence advances tumor growth, invasion, and metastasis [71].

The role of CSPGs In cancer progression and cell signaling is associated with both normal and pathological conditions since they regulate proliferation, apoptosis, migration, adhesion, invasion, and ECM assembly [71]. The research data showed that stromal versican regulated tumor growth by promoting angiogenesis. Several cancer cell lines have been used in the study, and tumors sourced from these cells had versican expression at high levels. However, in the case of Lewis lung carcinoma, both tumor and stroma had high-level versican expression [72]. Many studies have shown that some specific CSPGs, such as versican, are overexpressed in malignant tumors’ stroma [73,74]. On the contrary, recent research indicated that decorin, a type of CSPG, was a promising anticancer agent in osteosarcoma. The study showed that decorin affects cell motility due to the interaction of cells with matrix proteins. Although the inhibition of decorin expression by a decorin-specific siRNA did not affect MG-63 cell growth, it reduced cell motility [75]. Decorin (DS) is overexpressed in colorectal carcinoma, melanoma, osteosarcoma, and basal cell carcinoma; however, the expression of decorin decreased in other malignant tumors. This is due to the antiproliferative properties of decorin, which is a natural inhibitor of TGFβ, and the inhibition of this growth factor limits tumor bioavailability. Overall, CS functions in lipoprotein modification accumulation, inflammatory cell adhesion, chemokines binding, growth factor signaling, cell phenotype, and elastic fibers assembly, implicated in cancer progression and atherosclerosis development [65,76]. Additionally, the upregulation of some versican isoforms by TGFβ causes osteosarcoma cells to have aggressive behavior [77]. The chemical structure of GAGs, such as the presence of L-iduronic acid and sulfation type, affects cell growth of both osteosarcoma cells and osteoblastic lineage cells in a concentration-dependent manner. For instance, heparin significantly inhibited the proliferation of both normal osteoblasts and transformed osteoblastic cells at concentrations ≥1 uL/mL [78].

### 2.6. MicroRNAs and Protein Interactions

MicroRNAs (miRNAs) have been shown to function in tumorigenesis and tumor progression. For instance, miR-363 suppressed cancer in a variety of tumors, including gastric cancer [79], lung adenocarcinoma [80], and the metastasis of colorectal cancer [81]. In the case of osteosarcoma, miR-363 inhibited the proliferation, colony formation, and cell viability, promoting cell apoptosis and G1/S arrest in osteosarcoma [82]. Here, we will discuss the interaction of miRNAs and some proteins required for cellular processes.

#### 2.6.1. NOB1

NOB1, located on the human chromosome 16q22.1, expresses NOB1 protein in the nucleus of mammalian cells. RNA-binding protein NOB1 is a ribosome assembly factor that maintains cellular homeostasis by controlling protein degradation. Furthermore, NOB1 was shown to be essential for the cleavage of the 20S pre-rRNA into the mature 18S rRNA [83,84]. It is a part of a pre-40S ribosomal particle that is transported to the cytoplasm and subsequently cleaved at the 3′ end of mature 18S rRNA (D-site) [83]. In a study, a point mutation was designed in the NOB1 gene and the cells naturally expressing this protein failed to process the 20S pre-rRNA. Overall, this gene is a key factor for ribosomal biogenesis and affects RNAi and nonsense-mediated mRNA decay [84]. The relationship between miR-363 and NOB1 gene was investigated in osteosarcoma tissue specimens. The results illustrated that the downregulation of miR-353 upregulated NOB1 in osteosarcoma tissue specimens. The miR-363 overexpression had a detrimental effect on cell proliferation, migration, invasion, and epithelial-mesenchymal transition (EMT). The study emphasized that NOB1 could be a potential target mediated by miR-363 since the expression of NOB1was reversely correlated with the inhibitory effect of miR-363 on cell migration and invasiveness [85].

The studies indicated that the proteasome inhibiting drugs, including bortezomib and thiazole antibiotic thiostrepton, suppress the growth and induce apoptosis in osteosarcoma cell lines and xenografts, which is evident that NOB1 could be a potential therapeutic target for bone cancer [86,87,88,89].

The role of NOB1 gene silencing has been investigated in osteosarcoma, and the results demonstrated that the gene expression of NOB1 in human osteosarcoma cell lines, including SF-86, Saos-2, MG63, SW1353, and U2OS, was determined with Western blot analysis, in which Saos-2 and U2OS cells moderately expressed NOB1. This gene was knocked down by lentivirus-mediated shRNA to downregulate the NOB1 expression, and cell growth has been evaluated in MTT, colony-forming, and cell cycle assays. The results showed that the silencing of NOB1 significantly inhibited cell growth and caused osteosarcoma cells to arrest in the G2/M phase. This inhibition decreased cell migration while increasing the expression of tumor suppressor genes E-cadherin and β-catenin in U2OS cells. Both E-cadherin and β-catenin have been reported to associate with the metastatic progression of several types of cancer [86]. The potential role of the NOB1 gene was also explored in U251 and U87-MG higher-grade glioblastoma cell lines using a lentiviral vector (Lv-shNOB1). NOB1 expression and cellular localization were evaluated in 56 surgical glioma specimens.

NOB1 protein was localized in both the nucleus and cytoplasm in U251 cells, which was not associated with the malignancy level of glioma. Downregulation of NOB1 considerably inhibited cell proliferation and colony formation in human glioma cells. NOB1 silencing by Lv-shNOB1 induced G0/G1 phase arrest leading to cell apoptosis and suppressing cell migration in U251 and U87-MG cell lines. In conclusion, the regulation level of NOB1 may determine the aggressiveness of gliomas [84].

#### 2.6.2. HMGB1

High mobility group box 1 (HMGB1) is a highly conserved non-histone nuclear protein, which regulates transcription and is involved in the organization of DNA. It has pivotal functions in inflammation, cell differentiation, and tumor cell migration [90].

Furthermore, HMGB1 is located in the nucleus; however, it is translocated in cytoplasms to activate autophagy by binding to beclin1 [91]. The studies showed that HMGB1 might also induce metastasis and chemotherapy resistance in lung cancer [92], which supported that HMGB1-mediated autophagy led to chemotherapy resistance in osteosarcoma both in vitro and in vivo HMGB1 interaction with beclin1-PI3KC3 complex [93].

Moreover, doxorubicin, cisplatin, and methotrexate, the most used anticancer drugs, promoted HMGB1 expression in osteosarcoma cells, indicating the upregulation of HMGB1 during chemotherapy. The suppression of this gene caused sensitivity to chemotherapy, while the overexpression of HMGB1 increased resistance to chemotherapy in vitro. Knockdown of HMGB1 by shRNA increased the sensitivity of osteosarcoma cells to chemotherapy in NOD/SCID mice, in which autophagy decreased, whereas apoptosis increased in response to HMGB1-specific shRNA treatment [93]. The upregulation of autophagy was also observed in several cancer types, such as lymphoma, melanoma, leukemia, and breast cancer, in which it promoted or inhibited antitumor drug resistance. A study showed that the upregulation of HMGB1 promoted autophagy during chemotherapy in osteosarcoma cells, followed by drug resistance [94].

HMGB1 was expressed in bone and bone marrow at high levels. Bone marrow space closely interacts with the bone and immune cells, which play essential roles in cytokine intercellular signaling pathways. Outside the cells, HMGB1 acted as an immune response to pro-inflammatory cytokines when interacting with the receptor for advanced glycation end products (RAGE). Immunocytochemical analysis revealed that HMGB1 and RAGE were expressed in osteoblasts and osteoclasts [95].

Recently, new therapeutic approaches have displayed the modulation of HMGB1 by miRNAs in osteosarcoma cells, one of which used miR-505 to downregulate HMGB1 in human osteosarcoma. The study compared miRNA expression levels of 37 osteosarcoma samples and neighboring healthy cells. A total of 12 miRNAs were significantly upregulated, while 14 were downregulated, where miR-505 was involved. Decreased miR-505 levels were an indicator of poor clinical prognosis in osteosarcoma patients. In addition to that, HMGB1 mRNA levels were investigated, and the results showed that HMGB1 was overexpressed in osteosarcoma cells compared to the adjacent non-cancerous tissues. The gene expression of HMGB1 was negatively correlated with miR-505 levels in osteosarcoma samples. miR-505 suppressed the proliferation, migration, and invasion of MG63 cells in vitro, downregulating HMGB1 [96].

#### 2.6.3. MIF

Macrophage migration inhibitory factor (MIF), which regulates macrophage function, is a cytokine that modulates inflammation via counter-regulation of glucocorticoids and is involved in cell-mediated immunity, immunoregulation, cell proliferation, and tumorigenesis. MIF and JAB1 protein as a complex may have a role in integrin signaling pathways [97,98]. The human miR-451 is a type of miRNA located on chromosome 17q11.2. MiR-451 has acted as a tumor suppressor in many cancers, including nasopharyngeal carcinoma by targeting MIF [99] human glioma by downregulating the PI3K/Akt pathway through the calcium-binding protein 39 (CAB39) [100], and lung cancer by targeting ras-related protein 14 (RAB14) [101]. In a study, the expression of miR-451 in osteosarcoma tissue samples (hFOB and osteoblasts) and cell lines (U2OS and MG-63) were analyzed. MiR-451 was downregulated in osteosarcoma tissues and cell lines. Additionally, biological functions of miR-451 in osteosarcoma were explored by inducing miR-451 overexpression with miR-451 lentiviral vector transduction into U2OS cells, the results of which showed that the cell growth was attenuated in MiR-451 overexpressing osteosarcoma cells compared to the control group. This overexpression suppressed the proliferation and migration of osteosarcoma cells and inhibited the angiogenesis of the HUVEC cells. Cell apoptosis rates increased depending upon the upregulation of miR-451. In the in vivo part of the study, the growth inhibitory effect of miR-451 was investigated in nude mice in which LV-miR-451-U2OS cells were inoculated. Following three weeks, the overexpression of miR-451 suppressed tumor growth in nude mice. The critical point here was how miR-451 inhibited osteosarcoma cell growth and migration. For this purpose, the molecular targets of miR-451 were examined, in which MIF was found as a direct target of miR-451. Moreover, the silencing of MIF made similar changes in the proliferation, migration, and angiogenesis compared to the miR-451 mimic-transfected group. Overall, the upregulation of the miR-451 could inhibit proliferation and migration in osteosarcoma cells and induce apoptosis by the downregulation of MIF expression [102].

### 2.7. DNA-PKcs

DNA-dependent protein kinase catalytic subunit (DNA-PKcs) is a nuclear serine/threonine-protein kinase, a target of apoptosis protease. There is a correlation between the occurrence of apoptosis and the activity of DNA-PKcs, in which the activity of DNA-PKcs decreases following apoptosis. This may prevent the repair of doubled strand breaks to provide a smooth process for apoptosis. DNA-PK is required to be associated with DNA to have catalytic activity. This association is provided through Ku autoantigen, a DNA binding component [103].

The inhibition of DNA-PKcs caused tumor formation to decrease in vitro, reduced growth of human intracranial glioblastoma (GBM) xenografts in mice, sensitized the GBM xenografts to radiotherapy, and led to tumor regression in glioblastoma xenografts [104]. DNA-PKcs played a critical role in maintaining cell homeostasis. The effects of DNA-PKcs inhibition were investigated in the growth, migration, invasion, and apoptosis of osteosarcoma. A total of 57 osteosarcoma patients’ tumors and adjacent normal tissues were analyzed. The results indicated that the level of mRNA and protein expression of DNA-PKcs increased in osteosarcoma tissues compared to the neighboring healthy tissues. Furthermore, nine cases were with metastasis phase in osteosarcoma, and twelve cases were with lung metastasis among these patient samples. These samples with metastasis in osteosarcoma showed the highest mRNA expression level of DNA-PKcs compared to patients without metastasis. To confirm the effect of DNA-PKcs in osteosarcoma, DNA-PKcs was inhibited by siRNA-DNA-PKcs in MG63 cell lines, which illustrated a decreased mRNA and protein expression of DNA-PKcs. Silencing of this gene also decreased the expression of Cyclin D1, PCNA, and Bcl-2, whereas it increased Bax compared with control groups. Knockdown of DNA-PKcs suppressed the proliferation, migration, and invasion of MG-63 osteosarcoma cells; however, it promoted apoptosis of MG-63 [105]. Moreover, subcutaneous tissues of two nude mice inoculated with MG-63 cells after siRNA-DNA-PKcs treatment showed progress, in which tumor take and diameter were lower than in blank and siRNA control groups. The weight and size of tumor nodules on the lung surface were lower in the siRNA-DNA-PKcs than in the control groups.

### 2.8. GREM1

GREM1, bone morphogenetic protein antagonist, is a member of the BMP family, which regulates organogenesis, body patterning, and tissue differentiation [20]. Studies indicated that the overexpression of GREM1 was associated with the progression of tumors by facilitating invasiveness, including colorectal cancer [106], colon cancer [107], breast cancer [108], and mesothelioma [109]. Moreover, the silencing of GREM1 inhibited cell viability, migration, invasion, and EMT in glioma cells [110].

Based on the role of GREM1 in other cancers, the expression and function of GREM1 in osteosarcoma cells also was studied by Gu et al. Unlike other cancers mentioned before, GREM1 overexpression by lenti-GREM1 suppressed the cell viability, proliferation, invasion, and migration in U2OS and Saos-2 osteosarcoma cell lines. The GREM1 expression level was tested in hBMSC, hFOB1.19, Saos-2, MG63, and U2OS cell lines and the results showed the downregulation of GREM1 in Saos-2, MG63, and U2OS compared to others. Knockdown of GREM1 promoted the proliferation, invasion, and migration of U2OS and Saos-2 cells transfected with pLKO.1-GREM1 compared to the control group, although not affecting the apoptotic ability of these cell lines. In addition, the expression of matrix-degrading enzymes (metalloproteinase 2 and 9) and Id1 (inhibitor of DNA binding 1) was inhibited by GREM1 overexpression in U2OS, while overexpressed by the silencing of GREM1. The study elucidated that the upregulation of GREM1 could suppress the migration and invasion of HUVECs and inhibit endothelial cells from acquiring angiogenic ability. In in vivo experiments, nude mice were inoculated with U2OS cells to confirm the effects of GREM1 on osteosarcoma progression. Based on the measurement of tumor volumes, silencing of GREM1 increased the tumorigenesis of osteosarcoma cells, while the upregulation of this gene inhibited the proliferation of osteosarcoma in vivo [20].

## 3. Discussion

Osteosarcoma, which has a detrimental effect on the growth plate proliferation areas of long bones, is the most common bone cancer, commonly affecting people younger than 20 years old [111]. RNA interference (RNAi) is the post-transcriptional process, in which the gene expression leading to the disease has been regulated through the interference of mRNA where the genetic codes for the new protein synthesis are carried [12]. RNAi technologies provide a potential for treating genetic disorders, viral infections, and especially cancer, in which overexpressed or altered proteins may be regulated through RNAi. The critical prospect of this advance is to select correct sequences and synthesize appropriate RNAi [13]. This paper has reviewed studies using siRNA, shRNA, and miRNA in corporations with mostly lentiviral vectors applied to in vitro and in vivo, and potential molecular targets in osteosarcoma treatment.

As a cancer therapeutic agent, RNAi has demonstrated an effective treatment in phase I and II studies by silencing disease-causing genes [9,10,20]. Nevertheless, there have been several obstacles regarding safely and efficiently delivering synthesized RNAi types. Most researchers indicate that the efficient delivery of RNA-related gene products is an unmet challenge in many therapeutic approaches due to the instability of RNA. Therefore, carrying RNA gene products by viral vectors or nanoparticles is more reliable than using the naked RNAi approach [112]; however, these carriers still require optimizations to be able to use as therapeutics. Additionally, there are several studies focusing on the synthesis and chemical modification of siRNA to reduce side effects, in which siRNA nucleotides are modified chemically to develop chemically stable and efficacious RNAs, to increase target cell specificity, to reduce immune reaction and to decrease off-target effects [113,114]. Furthermore, the studies revealed that compared to other methods, viral vectors provide a broad tissue-specific tropism and high efficiency when the gene is silenced for a long term, in addition to the capability to encapsulate and deliver into the cell [13,115]. Lentiviral vectors provide a stable and long-term expression unlike other vectors [116]. The target specificity of the vector determines the siRNA concentration in the targeted tissue. Therefore, the disease-causing gene is blocked, and treatment efficiency increases. The off-targets in RNA delivery is another challenging field, and the studies showed that this might be overcome using specific receptors and peptides [115]. Another study showed that liposomes modified with YIGSR peptide targeting tumor cells prevented primary lung metastasis and angiogenesis [117]. Peptide selection to increase the affinity of a vector to the target tissue is crucial since it binds to a molecule of interest. In other words, the affinity of a peptide for the target increases therapeutic efficacy. Recent research indicated that phage display was a promising approach to choosing specific peptides with high affinity, increasing efficacy, and reducing off-target effects. Herein, the study also emphasized that three specific peptides bound to cultured chondrocytes were isolated with the help of the phage display method [118]. Taking all together, the approaches used in gene therapy should maximize the therapeutic effect.

Osteosarcoma has a complexity based on prognostic factors and risk groups, which adversely affects the development of new therapies [11]. Therefore, it is essential to note that next-generation sequencing of each osteosarcoma will guide effective treatment in pediatric groups. Guimaraes et al. showed that 50% of all osteosarcoma patients (84 in total) had somatic variants with TP53, MYC, CDK4, RB1, and PDGFRA genes. Moreover, MYC copy number variants were detected more frequently in tumors from patients under 10 years old (*p* = 0.023) [11]. Considering the age group of patients, early recognition and diagnosis improve the quality of life and protect patients from osteoblastic lineage mutations. It was shown that osteosarcoma cells directly induce bone marrow mesenchymal stem cells to cancer-associated fibroblasts in vitro, and Notch and Akt signaling pathways regulate this differentiation [3]. These data still need to be explored in mice and then human models to prevent metastasis in different tissues. Studies confirmed the efficiency of individualized therapies both in vitro and in vivo. For example, the investigation of the USP1 gene on 30 osteosarcoma patients showed that 26 had similar properties resulting from the overexpression of the USP1 gene while four were different, although all patients were already diagnosed with osteosarcoma. However, the knockdown of the USP1 gene by a lentiviral vector carrying USP1-specific shRNA suppressed mRNA expression and the translational process of this gene in U2OS cells [9]. Furthermore, the dysregulation of long non-coding RNAs (LncRNAs) affects cancer progression due to their adverse effect on cell growth, metastasis, apoptosis, and differentiation. Knockdown of GHET1, one of the lncRNAs upregulated in osteosarcoma, by naked siRNA improved tumor growth and metastasis in vivo [119].

The regulation of cancer-related genes will change based on patient physiology. Instead of administering the same approach to all patients, individuals should be evaluated for a tailor-made medicine, since tumor cells are unique for each patient. For this purpose, molecular methods such as polymerase chain reaction and sequencing are excellent approaches for early diagnosis and treatment combined with histological analysis. The use of the RNAi approach with a well-designed delivery option may provide effective treatment on the target site.

## Figures and Tables

**Figure 1 ijms-23-12583-f001:**
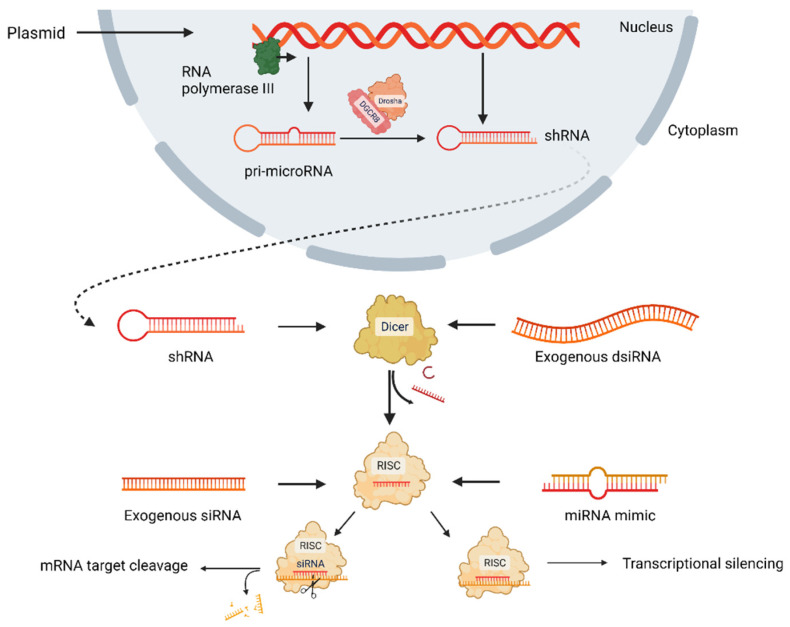
RNA interference pathways. The upstream process starts with short hairpin RNA (shRNA) or pri-microRNA (pri-miRNA) addition, which requires nuclear processing, followed by transportation into the cytoplasm. Upon arrival, shRNA and Dicer substrate RNA (dsiRNA) are digested into short interfering RNA (siRNA) or microRNA (miRNA). To produce siRNAs, the dicer enzyme cleaves double-stranded RNA (dsRNA) sequences into small double-stranded siRNA, and RISC elements transport them to mRNA targets that recognize guide-stranded RNA (gsRNA), which results in mRNA cleavage and degradation. In the process of microRNA (miRNA) delivered into cytoplasm, RISC binds this target sequence, which results in translational repression and successive segregation into p-bodies for degradation [12,13,14,18]. Adapted from “RNAi mechanism”, by BioRender.com (2022). Retrieved from https://app.biorender.com/biorender-templates.

**Figure 2 ijms-23-12583-f002:**
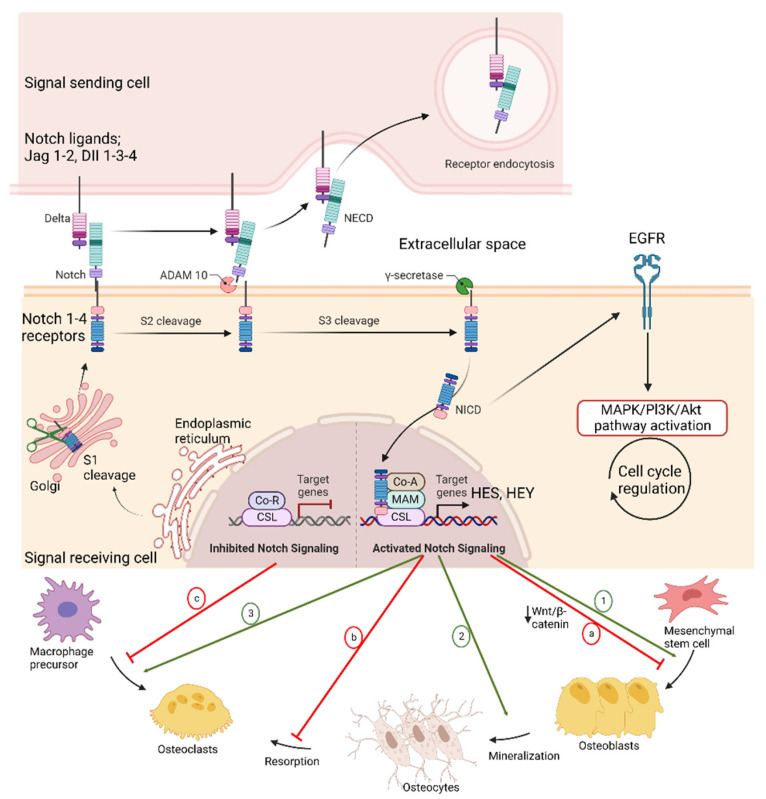
Regulation of Notch signaling. Notch inactive signal peptide precursor is synthesized in endoplasmic reticulum and moves to Golgi. After S1 cleavage process by furin-like convertases, it is translocated into the cell membrane. Following signal transduction from adjacent cell, S2 cleavage occurs via the disintegrins and metalloproteinases (ADAM). Truncated notch fragment is further processed by S3 and S4 cleavage, and thus, Notch intracellular domain (NICD) forms. NICD is the active form of the Notch receptor, which can direct transcriptional activity by entering the nucleus [53]. Notch signaling activation stimulates osteogenic differentiation and bone mineralization while inhibiting osteogenesis by suppressing Wnt/β-catenin signaling. Notch inhibition by a signaling inhibitor can enhance osteogenesis (1, 2 and a). Activated Notch signaling in osteocytes can suppress bone resorption and increase bone density by reducing sclerostin and dickkopf Wnt signaling pathway inhibitor 1 while upregulating Wnt signaling (b). Notch 1-3 deficiency directly promotes osteoclastic differentiation, and Notch1 deficiency can promote this process indirectly by decreasing OPG and RANKL ratio. The Notch1/Jagged1 suppresses osteoclastogenesis, whereas the Notch2/Delta-like (DLL) 1 activates. Notch activation affects osteoclast formation since it strongly inhibits M-CSF (3 and c) [54]. Adapted from “Notch Signaling Pathway”, by BioRender.com (2022). Retrieved from https://app.biorender.com/biorender-templates.

## Data Availability

Not Applicable.

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
