# Peer review of "Regulation of Molecular Targets in Osteosarcoma Treatment"

_ijms, 2022, doi:10.3390/ijms232012583_

Round 1
Reviewer 1 Report
The review has a broad approach to the studies carried out looking for molecular targets for the treatment of OS. The authors also propose the use of RNAi technology to direct the therapy to these targets. However, the text in different sections and some figures must be revised to reach the quality for publication. I Attached in the file the comments where the changes are suggested.

Author Response
Reviewer 1
Thank you very much for your positive comments. We have revised the manuscript according to your critical comments. We hope this version meets your comments.
Line 31-33: In this paragraph, the authors should explain why the revision is focused only on osteosarcoma among the other bone cancer mentioned
Response: We have included, “Osteosarcoma is the eighth most common childhood malignancy, comprising 2.4% of pediatric cancers, including leukemia (30%), brain and nervous system cancers (22.3%), neuroblastoma (7.3%), Wilms tumor (5.6%), Non-Hodkin lymphoma (4.5%), rhabdomyosarcoma (3.1%), retinoblastoma (2.8%), and Ewing sarcoma (1.4%) [1,2]. Osteosarcoma, which has a bimodal age distribution, is considered to result from the cells mutated in osteoblastic lineage depending on their susceptibility during osteoblastic differentiation [1]. Osteosarcoma cells from osteoblastic lineage induce mesenchymal bone marrow cells into cancer-associated cells [3]. Osteosarcoma affects pediatric groups (mostly 10-14 years of age) accounted for the first osteosarcoma peak and older adulthood (older than 65 years of age) in the second osteosarcoma peak [2]. The incidence of osteosarcoma is higher in males (5.4 per million persons per year) than in females (4.0 per million persons per year). Among populations, black people have the highest incidence of osteosarcoma (6.8 per million persons per year), followed by Hispanics (6.5 per million) and Whites (4.6 per million). The most common site of osteosarcoma is commonly in the long bones near the metaphyseal growth plates; the femur (42%; 75% of tumors in the distal femur), the tibia (19%; 80% of tumors in the proximal tibia), and the humerus (10%; 90% of tumors in the proximal humerus). Additionally, osteosarcoma may occur in the skull or jaw (8%) and the pelvis (8%). Regarding mortality rates, bone and joint malignancies accounted for 8.9% of all childhood and adolescent cancer deaths [2].” in lines 32-53; 55-62.
Line 34-35: The graph in figure 1 is not related to the mention in the text "Studies have shown that somatic copy number changes, along with recurrent point mutations, often lead to the development of osteosarcoma."
Response: We have removed Figure 1 and included the following text (lines 59-62).
“Osteosarcoma not only occurs with mutations in primary mesenchymal stem cells but also results from cancer cells migrating from primary tumor sites through blood circulation (Fig. 1) [7].”
Line 54-57: The figure should be eliminated or changed to another that is related to what is mentioned in the text about the genetic and molecular changes that lead to the development of osteosarcoma.
Response: We have removed Figure 1 and added the genetic and molecular changes in the text (lines 54-62).
Line 58-59: To explain better (for Figure 2)
Response: We have included, “In the RNAi approach, long mRNA transcripts are diced by the Dicer enzyme to create short double-stranded RNAs (dsRNAs), and these dsRNAs are loaded onto RNA-induced silencing complex (RISC) comprising Argonaute and transactivation response RNA-binding protein. After cleaving one strand, the guide strand is paired with its complementary mRNA target via RISC. Following binding, the mRNA is silenced via two pathways: RNase-mediated degradation or translational repression [13–16]. RNAi approach has four types: small interfering RNA (siRNA) and microRNA (miRNA) mimics, short hairpin RNAs (shRNAs), and Dicer substrate RNAs (dsiRNAs), which are non-coding RNAs (ncRNAs) and post-transcriptionally regulate protein synthesis. siRNA is a short double-stranded ncRNA, having 20-25 nucleotides loaded onto RISC, and they degrade and cleave mRNAs containing specific nucleotide sequences before being translated. miRNA having 19-25 nucleotides is another double-stranded ncRNA loaded onto RISC to regulate gene at the posttranscriptional level by targeting mRNA sequence [17,18]. shRNA requires nuclear processing and is most upstream, while dsiRNA requires Dicer processing. siRNA and miRNA pathways are the most direct; however, their silencing outcome is different because siRNAs are 100% complementary of mRNA target sequences, but miRNAs are not. miRNAs induce translational repression, while siRNAs induce Argonaute2-mediated degradation [13]. All four types of RNAi approaches are delivered by inserting into either nanoparticles or viral vectors, or alone currently in clinical trials [13]. As a part of the natural cellular process, siRNAs are administered directly into cells, tissue, or organisms; however, shRNAs that provide siRNA are transported into cells through a vector, and siRNA is expressed when shRNA is processed [19].” in lines 88-110 (Figure 2 is now Figure 1).
Line 69: The authors should describe in a figure legend the mechanism of RNAi approach
Response: We have included, “Figure 1. RNA interference pathways. The upstream process starts with short hairpin RNA (shRNA) or pri-microRNA (pri-miRNA) addition, which requires nuclear processing, followed by transportation into the cytoplasm. Upon arrival, shRNA and Dicer substrate RNA (dsiRNA) are digested into short interfering RNA (siRNA) or microRNA (miRNA). To produce siRNAs, the dicer enzyme cleaves double-stranded RNA (dsRNA) sequences into small double-stranded siRNA, and RISC elements transport them to mRNA targets which recognize guide-stranded RNA (gsRNA), which results in mRNA cleavage and degradation. In the process of microRNA (miRNA) delivered into cytoplasm, RISC binds this target sequence, which results in translational repression and successive segregation into p-bodies for degradation [13–15,19].” in lines 118-127.
Line 71: Since the authors propose the use of RNAi technology, in this section they should discuss and connect how each molecular target in OS will be regulated by RNAi approachs
Response: We have discussed and connected RNAi approach use in the context of osteosarcoma.
Line 598: In the discussion section, the authors should be able to draw their conclusions about why to propose and even more so the advantages of using ANN and exclusive technology for the OS. Also discuss how it looks in the future, near or far, the clinical use of these molecular targets and their regulation by RNAi in the treatment of OS and even more so in metastatic disease.
Response: We have revised the Discussion Section in lines 658-661; 664-668; 672-681; 698-709; 717-719; 721-722; 725-726 as suggested.
Author Response
Reviewer 2
The review manuscript by Celik and colleagues summarizes some dysregulated genetic pathways found in bone tumors and discusses the potential use of new technologies to interfere with gene regulation and protein expression for improving current therapies. In principle, the topic may fit into the scope of the journal. The review includes some useful information, but also misses important aspects of molecular profiling of subsets of osteosarcomas, or even mentioning Rb1 tumor suppressor gene involved in 30% of those tumors. There also seems to be one major issue with figure 1, but also some minor issues and wrong references, which perhaps should be addressed with major revisions of the manuscript. Adding a table listing the crucial targets may also help a lot. Abbreviations should be explained once in the text (siRNA, shRNA, and gsRNA – abbreviations used, but never explained), but not repeatedly, like RNAi (even in the discussion). Overall, the manuscript didn’t convince the reviewer of anything new and exciting specifically in the field of osteosarcoma to accelerate such a review.
Thanks a lot for your critical comments. We have revised the manuscript according to your suggestions. We hope this version satisfies your comments.
Specific remarks:
1) Not all references appear to be ideally chosen, one checked is even wrong. Please check all references.
Response: Thank you very much for your critical suggestions. We have checked the refs and put the correct refs in the appropriate place.
Examples:
Reference 1 – not ideal, not necessary here to link just to a webpage, there should be solid literature around; suggestion: cite real literature, which will be easily findable in the future
Response: We have updated it according to the suggestion.
Ref. 2. Not an ideal, original citation for the point. Suggestion: replace with better reference for the point “cell of origin”.
Response: We have replaced it with the original ref.
Ref. 3. Something is VERY WRONG with this reference: Title “Recurrent Somatic Structural Variations Contribute to Tumorigenesis in Pediatric Osteosarcoma” refers to different paper from other authors and from 2014 in Cell Rep. The listed authors Mirrielees, Crofford, Lin et al. published on rheumatoid arthritis in another journal, other year.
Response: We are very sorry that we misplaced the ref. We have corrected the ref.
Line 31: should be mentioned somewhere, how common/rare these types of cancers really are, with reference.
Response: We have included, “Osteosarcoma is the eighth most common childhood malignancy, comprising 2.4% of pediatric cancers, including leukemia (30%), brain and nervous system cancers (22.3%), neuroblastoma (7.3%), Wilms tumor (5.6%), Non-Hodkin lymphoma (4.5%), rhabdomyosarcoma (3.1%), retinoblastoma (2.8%), and Ewing sarcoma (1.4%) [1,2]. Osteosarcoma, which has a bimodal age distribution, is considered to result from the cells mutated in osteoblastic lineage depending on their susceptibility during osteoblastic differentiation [1]. Osteosarcoma cells from osteoblastic lineage induce mesenchymal bone marrow cells to cancer-associated cells [3]. Osteosarcoma affects pediatric groups (mostly 10-14 years of age) accounted for the first osteosarcoma peak, and older adulthood (older than 65 years of age) in the second osteosarcoma peak [2]. The incidence of osteosarcoma is higher in males (5.4 per million persons per year) than in females (4.0 per million persons per year). Among populations, black people have the highest incidence of osteosarcoma (6.8 per million persons per year), followed by Hispanics (6.5 per million) and Whites (4.6 per million). The most common site of osteosarcoma is commonly in the long bones near the metaphyseal growth plates; the femur (42%; 75% of tumors in the distal femur), the tibia (19%; 80% of tumors in the proximal tibia), and the humerus (10%; 90% of tumors in the proximal humerus). Additionally, osteosarcoma may occur in the skull or jaw (8%) and the pelvis (8%). Regarding mortality rates, bone and joint malignancies accounted for 8.9% of all childhood and adolescent cancer deaths [2]. Studies have shown that somatic copy number changes, along with recurrent point mutations, often lead to the development of osteosarcoma [4,5]. In recent years, the prevalence of germline mutations among pediatric cancer patients has reached 7.9%, which is associated with several cancer predisposition disorders such as autosomal dominant Li-Fraumeni and Hereditary Retinoblastoma, and autosomal recessive Werner, Bloom, Rothmud-Thompson, and Rapadilino syndromes [6].” in lines 32-53; 55-62.
Line 42-44: please add reference? (or rephrase)
Response: We have removed lines 42-44.
2) Line 54: Figure 1 is –out of the blue- about primary tumors from breast, liver, brain (!), supposedly metastasizing to the bone (which is almost never the case for brain tumors), but this has really NOTHING to do with the topic osteosarcoma. Suggestion: remove or replace figure with one better related to the topic – or explain better the relation to the topic osteosarcoma. Figure 1 doesn’t explain the abbreviations, like RANKL, RANK.
Response: We have removed Figure 1.
3) Figure 2: include legend, explaining Drosha, Dicer, RISC. Suggestion: replace “nucleus” with “Nucleus” to adapt better to other figure.
Response: We have updated Figure 2 and included the legend as follows. (Figure 2 is now Figure 1)
“Figure 1. RNA interference pathways. The upstream process starts with short hairpin RNA (shRNA) or pri-microRNA (pri-miRNA) addition, which requires nuclear processing, followed by transportation into the cytoplasm. Upon arrival, shRNA and Dicer substrate RNA (dsiRNA) are digested into short interfering RNA (siRNA) or microRNA (miRNA). To produce siRNAs, the dicer enzyme cleaves double-stranded RNA (dsRNA) sequences into small double-stranded siRNA, and RISC elements transport them to mRNA targets which recognize guide-stranded RNA (gsRNA), which results in mRNA cleavage and degradation. In the process of microRNA (miRNA) delivered into cytoplasm, RISC binds this target sequence, which results in translational repression and successive segregation into p-bodies for degradation [13–15,19].” in lines 118-127.
4) Line 265-267: suggestion: rephrase, Unclear
Response: We have revised it.
5) Line 286 (Figure 3) suggestion: think about replacing “γ secretase” in the Figure with “γ-Secretase” (keep the symbol for “gamma” (γ) in front of “–Secretase”.
Response: We have replaced “γ secretase” with “γ-secretase.”
6) Please check, if the writing is correct: “ γ secretase “, or if “γ-secretase” (or “γ-Secretase” ! in the figures) may be better.
Response: We have replaced “γ secretase” with “γ-secretase” through the text and figures.
7) Line 599ff: Discussion should be rewritten parts of the discussion may fit to earlier segments; explaining siRNA (line 609) and introducing argonaute 2 (Ago 2) in the discussion – or mentioning gsRNA in the discussion without explaining – should be improved, or deleted
Response: We have revised Discussion Section in lines 658-661; 664-668; 672-681; 698-709; 717-719; 721-722; 725-726 as suggested.

Round 2
Reviewer 1 Report
The authors have responded correctly to all suggestions. Therefore, I consider that the review has been improved. I understand that figure 1 has been completely removed (although it is still in the text), if so and can be published as is presented in the latest version.
Author Response
Thank you very much for your positive comments. We have revised the manuscript according to your critical comments. We hope this version meets your comments.
Response: We have removed Figure 1 as suggested.
Reviewer 2 Report
In the revised version the authors addressed the critical suggestions and improved it significantly in many ways: e.g. the deletion of the totally misleading original Figure 1 (mixing up bone metastasis of any cancer with osteosarcoma genesis), improvement of English, as well as adding new references and data on osteosarcoma.
The reviewer suggests now only minor modifications to make it publishable as a general review, although it doesn't seem to be really necessary for osteosarcomas: check again to avoid an obviously continuing and repeated misunderstanding between “bone metastasis” (of any cancer) and “osteosarcoma” (originating in the bone, but metastasizing anywhere else, like mainly into the lungs, which causes death, but only occasionally also into the bone).
Overall, this may be publishablefand interesting for readers outside of pathology - with minor deletions or corrections.
Specific remarks:
11) Line 65: unclear (or wrong): osteosarcoma is the primary tumor this manuscript is about. Unfortunately, as with the first version of the manuscript (old/now deleted Figure 1), there seem to be a continuing lack of understanding the difference between osteosarcoma and other cancers (breast, colon, prostate etc.) just metastasizing to the bone. The reference [7] is on bone metastasis (of all cancers) - and includes only 3 references on osteosarcoma.
22) Line 681: again: Osteosarcoma arises in the bone, but mainly metastasizes to the lung and most other organs, only occasionally to the bones! Therefore, it appears to be wrong to discuss bone metastasis - without discussing the main reason of death: metastasis to other organs, like lungs.
33) Line 685: should probably be deleted
Author Response
Thanks a lot for your critical comments. We have revised the manuscript according to your suggestions. We hope this version satisfies your comments.
In the revised version the authors addressed the critical suggestions and improved it significantly in many ways: e.g. the deletion of the totally misleading original Figure 1 (mixing up bone metastasis of any cancer with osteosarcoma genesis), improvement of English, as well as adding new references and data on osteosarcoma.
The reviewer suggests now only minor modifications to make it publishable as a general review, although it doesn't seem to be really necessary for osteosarcomas: check again to avoid an obviously continuing and repeated misunderstanding between “bone metastasis” (of any cancer) and “osteosarcoma” (originating in the bone, but metastasizing anywhere else, like mainly into the lungs, which causes death, but only occasionally also into the bone).
Response: We are sorry for confusing the reviewer. Misunderstanding of bone metastasis concept has been corrected and removed from the text. Our main goal here is to discuss osteosarcoma occurrence, not bone metastasis.
Overall, this may be publishable and interesting for readers outside of pathology - with minor deletions or corrections.
Specific remarks:
1) Line 65: unclear (or wrong): osteosarcoma is the primary tumor this manuscript is about. Unfortunately, as with the first version of the manuscript (old/now deleted Figure 1), there seem to be a continuing lack of understanding the difference between osteosarcoma and other cancers (breast, colon, prostate etc.) just metastasizing to the bone. The reference [7] is on bone metastasis (of all cancers) - and includes only 3 references on osteosarcoma.
Response: We have removed the related sentences and references from the text.
2) Line 681: again: Osteosarcoma arises in the bone, but mainly metastasizes to the lung and most other organs, only occasionally to the bones! Therefore, it appears to be wrong to discuss bone metastasis - without discussing the main reason of death: metastasis to other organs, like lungs.
Response: We have removed the related sentences and references from the text.
3) Line 685: should probably be deleted
Response: We have removed the related sentences.